# A Mechanics Analysis of Carbon Fiber Plain-Woven Thermoset Prepreg during Forming Process Considering Temperature Effect

**DOI:** 10.3390/polym14132618

**Published:** 2022-06-28

**Authors:** Jialiang Qi, Lun Li, Yiqi Wang, Hang Gao

**Affiliations:** Key Laboratory for Precision and Non-Traditional Machining Technology of Ministry of Education, School of Mechanical Engineering, Dalian University of Technology, Dalian 116024, China; qijialiang@mail.dlut.edu.cn (J.Q.); li_lun@mail.dlut.edu.cn (L.L.); gaohang@dlut.edu.cn (H.G.)

**Keywords:** carbon fiber plain-woven thermoset prepreg, finite element method, continuum model, polymer resin characteristics, preforming quality

## Abstract

The preforming quality of carbon fiber plain-woven thermoset prepreg (CFPWTP) is critical to the performance of composite aerospace parts. The deformation ability of the CFPWTP material during preforming is affected by both the fabric woven structure and the resin viscosity, which is different from the dry textile material. Incorrect temperature parameters can enlarge the resin’s viscosity, and high viscosity can inhibit fiber deformation and cause defects. This study proposes an equivalent continuum mechanics model considering its temperature–force behavior. Picture frame tests and axial tensile tests at 15 °C, 30 °C, and 45 °C are conducted to obtain the temperature–stress–strain constitutional equations. By Taylor’s expansion formula and surface fitting method, the constitutive modulus of the material is obtained. Consequently, a saddle-shaped forming simulation is carried out, which is later validated by experiments. Results show that the accuracy of the predicted model is high, with 0.9% of width error and 5.1% of length error separately. Besides, the predicted wrinkles are consistent with the test in fold position and in deformation trend under different temperatures.

## 1. Introduction

Advanced composite materials significantly improve performance and reduce aviation parts’ weight and cost. Composites, especially the carbon fiber plain-woven thermoset prepreg (CFPWTP), have been increasingly accepted in the aerospace manufacturing industry because of their light weight, high strength, corrosion resistance and formability [1]. When designing composite parts with CFPWTP material, this laminated material usually needs to be preformed layer by layer on the nondevelopable mold surface [2]. At this time, the material will endure great in-plane tensile, compression or shear deformation, which is mainly presented as the interfiber slippage and rotation. Some fiber bundles will be stretched or stacked together if the fiber movement is hindered, and local materials will be bridged or wrinkled [3,4]. Without the correct operation, these defects will seriously damage the aerodynamic shape or interlayer strength of products. Common treatments [5,6,7] include stretching, pressing the defect area, heating or redesigning the cutting size. However, arranging these processes requires firstly determining the deformation characteristics of the CFPWTP material.

The deformation of CFPWTP material is affected by both structure and components. From the perspective of structure, CFPWTP is the product of weaved carbon fiber submerged in resin, which leads to the complex contact interface between these two components. As for the component, CFPWTP inherits the excellent axial mechanical properties and low shearing stiffness [8] of dry woven fabrics as well as the resin’s polymer characteristics such as viscoelasticity. Researches [9,10] show that the resin matrix can significantly affect the interply shear resistance of woven fabrics.

In order to obtain material property, many efforts have been devoted through the last decades. Boisse et al. [11,12] and Cao et al. [13] have systematically studied the mechanics of glass fibers with many test methods. They have revealed the characteristic of dry glass fabric materials. Bilisik et al. [14] found that different length–width ratios of samples and the number of pulling fibers influence the shear results significantly. Nosrat-Nezami et al. [15] improved the design of the original picture frame (PF) device and carried out shear tests under different pre-tension conditions. The results show that the shear modulus increases nonlinearly and monotonically with the pre-tension load. Naito et al. [16] experimentally evaluated resin flow mechanism change in prepreg processing during cure including the gelation stage. Wang et al. [17] did work to investigate the interply slipping behaviors to avoid defects such as wrinkling during the manufacturing of large-scale composite parts in hot diaphragm forming.

Besides, some researchers have developed a series of nonlinear constitutive models [18,19] to characterize this material. Peng and Cao [20,21] built the non-orthogonal constitutive model for woven composite. Charmetant et al. [22] established a hyperelastic material model based on continuum mechanics. In addition, based on the woven unit cell, Boisse et al. [23,24,25] created a hyperelastic model which considers the effects of tensile and shear of fiber bundles in mesoscale. Haghi Kashani [26] studied the impact of tension–shear coupling in his model. Chen [27] studied the noncrimp fabrics (NCF) material model and the process optimization method of eliminating wrinkles through a genetic algorithm. Komeili [28] analyzed the fabric model from a multi-scale perspective. In addition, based on the forming theory, three kinds of models have been developed: the dynamic model [29], the discrete model [30] and the continuum model [31]. However, the dynamic and discrete models are not suitable for analyzing the whole formation because of the enormous calculation and lack of material property. Many researchers use the continuous model to simulate the woven fabric composite because it can be embedded in the finite element method (FEM) software [32].

The wrinkle is a typical defect of woven composites [33,34]. Generally, there are two types of wrinkles. 1. In-plane fracture [35]: the material’s warp and weft fibers reach the shear locking angle [36,37,38,39], resulting in the inability of the material to continue shear deformation, resulting in the tow buckling and fiber fracture. 2. Out-of-plane bending [40]: the material does not necessarily reach the shear locking angle but folds due to material compression.

There has already been much research on fabrics [41,42,43,44], but few models consider the effect of temperature change on the deformation capacity of materials. For example, it is difficult to move or shear the tow from the woven structure at low temperature due to resin consolidation, while at high temperature when the resin is melted, the above situation is the opposite. Therefore, to characterize the CFPWTP materials, it is necessary to consider both resin’s and fiber’s properties.

This study establishes an equivalent model of CFPWTP material based on the continuum mechanics for industrial forming processes. The new model is developed from the dry woven-fabric equivalent model but considers the resin’s characteristics variety. Then, the picture frame tests and axial tensile tests are carried out at three temperatures to obtain the model’s mechanical-temperature characteristic curves. With Taylor’s expansion method and fitting of the above results, the constitutive matrix conducive to simulation is obtained. The forming experiments are prepared and compared to examine the simulation’s accuracy, and then the wrinkle prediction and the comparison are arranged. The conclusions are prepared in the final section. The structure can be seen in Figure 1. 

## 2. An Equivalent Continuum Model of CFPWTP Material

This section will introduce the model in three steps. 1. Build a dry fabric model based on the continuum mechanics. 2. Analyze the resin’s temperature-viscosity characteristics. 3. Construct the model for characterizing CFPWTP materials.

Before the first step, it is necessary to analyze plain-woven prepreg characteristics for the model building. For example, see Table 1, the parameters of the Toray^®^ T1100G datasheet of the plain-woven carbon fiber prepreg are offered here. Firstly, through observation, it can be found that the prepreg thickness is very thin. If the composite size is 500 mm × 500 mm, the ratio of length/thickness or width/thickness exceeds 1000. Hence, it is acceptable to set aside the thickness of the model. Because plain weave fabrics 0°and 90° tensile modules are nearing, it is reasonable to consider that the mechanical parameters of the warp and the weft are the same. 

### 2.1. The Model of Plain-Woven Fabric

#### 2.1.1. The Material Mechanic Analysis

Researchers treat the plain-weave fabric material as macrolevel periodic materials consisting of a series of mesoscale patterns. This mesoscale pattern is called the representative unit cell (RUC) [46]. For the plain-woven dry fabric, a constitutive model [47] is created as follows. Firstly, acquire the material’s characteristic matrix. Secondly, establish the transforming tensor matrix of the global orthogonal coordinate and non-orthogonal coordinate. Thirdly, replace the RUC with the square plate element of FEM, see Figure 2. 

Because the thickness is negligible, the material element can be regarded as a thin plate. Based on the mechanics of materials and FEM, the thin plate can be treated as a shell element which has the following mechanics matrix $\left[\tilde{D}(\varepsilon )\right]$:(1)$$\left\{d{\tilde{\delta }}^{}\right\}=\left[\tilde{D}(\varepsilon )\right]\left\{d\tilde{\varepsilon }\right\}$$
(2)$$\left[\tilde{D}\right]=\left[\tilde{D}\left(\varepsilon \right)\right]=\left[\begin{array}{cc} {\tilde{D}}^{11}\left(\varepsilon \right) & {\tilde{D}}^{12}\left(\varepsilon \right)\\ \begin{array}{l} {\tilde{D}}^{21}\left(\varepsilon \right)\\ 0 \end{array} & \begin{array}{l} {\tilde{D}}^{22}\left(\varepsilon \right)\\ 0 \end{array} \end{array}\begin{array}{c} 0\\ 0\\ {\tilde{D}}^{33}\left(\varepsilon \right) \end{array}\right]=\left[\begin{array}{cc} {\tilde{D}}^{\mathrm{axial}}\left(\varepsilon_{\mathrm{axial}11}\right) & {\tilde{D}}^{\mathrm{trans}}\left(\varepsilon_{\mathrm{trans}12}\right)\\ \begin{array}{l} {\tilde{D}}^{\mathrm{trans}}\left(\varepsilon_{\mathrm{trans}21}\right)\\ 0 \end{array} & \begin{array}{l} {\tilde{D}}^{\mathrm{axial}}\left(\varepsilon_{\mathrm{axial}22}\right)\\ 0 \end{array} \end{array}\begin{array}{c} 0\\ 0\\ {\tilde{D}}^{\mathrm{shear}}\left(\varepsilon_{\mathrm{shear}}\right) \end{array}\right]$$
where $\left[{\tilde{D}}^{11}\left(\varepsilon \right)\right]$, $\left[{\tilde{D}}^{12}\left(\varepsilon \right)\right]$ and $\left[{\tilde{D}}^{33}\left(\varepsilon \right)\right]$ stand for the axial tensile modulus, the transverse modulus and the shear modulus. $\left\{d\tilde{\delta }\right\}$ and $\left\{d\tilde{\varepsilon }\right\}$ mean the stress and strain matrix. Besides, for simplicity, the warp direction is defined as the 1 direction and the weft corresponds to the 2 direction.

Because the fiber bundle is too soft, it is difficult to directly acquire the exact value $\left[{\tilde{D}}^{12}\left(\varepsilon \right)\right]$ from the test. Referring to the provided method [48] that ε_12_ has little effect on the deformation, $\left[D^{\mathrm{trans}}\left(\varepsilon_{\mathrm{trans}}\right)\right]$ is considered equal to Const × ε_11_, where Const is 0.02.

Based on the above the analysis, the $\left[{\tilde{D}}^{\mathrm{axial}}\left(\varepsilon_{\mathrm{axial}11}\right)\right]$, $\left[{\tilde{D}}^{\mathrm{trans}}\left(\varepsilon_{\mathrm{trans}12}\right)\right]$ and $\left[{\tilde{D}}^{\mathrm{shear}}\left(\varepsilon_{\mathrm{shear}}\right)\right]$ can be solved:(3)$${[D}^{\mathrm{axial}}\left(\varepsilon_{\mathrm{axial}11}\right)]=\left[{\tilde{D}}^{11}\left(\varepsilon \right)\right]{=\mathrm{Function}(\varepsilon }_{\mathrm{axial}}{,\sigma }_{\mathrm{axial}})$$
(4)$${[D}^{\mathrm{shear}}\left(\varepsilon_{\mathrm{shear}}\right)]=\left[{\tilde{D}}^{33}\left(\varepsilon \right)\right]=\mathrm{Function}(\varepsilon_{\mathrm{shear}},\sigma_{\mathrm{shear}})$$
(5)$${[D}^{\mathrm{trans}}\left(\varepsilon_{\mathrm{trans}12}\right)]=\left[{\tilde{D}}^{12}\left(\varepsilon \right)\right]=\mathrm{Const}\times {\tilde{D}}^{11}\left(\varepsilon \right)$$
where the function comes from the PF test and axial tensile test of Figure 3. 

To detect the function’s specific expression, it is necessary to establish the relationship between the strain, the stress of the model and the displacement, the force from the tests. The following equations are given in combination with the dimension of the material and device. Specifically, for shear stress δ_Shear_ and the strain ε_Shear_,
(6)$$\left\{ \begin{array}{l} F_{\mathrm{shear}}=F/(\frac{\sqrt{2}L_{\mathrm{arm}}+\mathrm{Path}}{L_{\mathrm{arm}}})\\ \delta_{\mathrm{shear}}{=F}_{\mathrm{shear}}/\left(L_{\mathrm{mat}}\times T_{\mathrm{mat}}\right) \end{array}\right.$$
(7)$$\varepsilon_{\mathrm{shear}}=90-2\arccos\;\left(\frac{\sqrt{2}L_{\mathrm{arm}}+\mathrm{Path}}{{2L}_{\mathrm{arm}}}\right)$$
where F is the tensile force directly from the test machine, F_shear_ is the solved shear force acting on the material, L_mat_ is the effective length of the cross-shape material, T_mat_ is the thickness of the material, L_arm_ is the arm length of the articulated arm of the PF, and the Path is the displacement of the machine clamp.

For tensile stress δ_Axial_ and the strain ε_Axial_, they could be solved from the following equations
(8)$$\delta_{\mathrm{axial}}=F/\left(W_{\mathrm{mat}}\times T_{\mathrm{mat}}\right)$$
(9)$$\varepsilon_{\mathrm{axial}}=\frac{\mathrm{Path}}{L_{\mathrm{mat}2}}$$
where F is the tensile force, W_mat_ is the effective width of the material, T_mat_ is the thickness of the material, L_mat2_ is the effective length of the tensile material, and the Path is the displacement of the machine clamp.

#### 2.1.2. The Orthogonal Coordinate Transforming

Based on the similarity between the square plate element and the RUC pattern, the edge strain vectors of plate elements are used to represent the warp and weft fiber of RUC. Thus, the strain vector of the non-orthogonal coordinate system needs to be transformed into the global orthogonal coordinate for the later FEM analysis. With tensor and continuum mechanics, the relation between strain dε from the orthogonal coordinate and its corresponding $d\tilde{\varepsilon }$ from the arbitrary non-orthogonal coordinate is given by
(10)$$\left\{d\tilde{\varepsilon }\right\}=\left\{\begin{array}{l} {d\tilde{\varepsilon }}_{11}\\ {d\tilde{\varepsilon }}_{22}\\ {d\tilde{\gamma }}_{12} \end{array}\right\}=\left[\begin{array}{ccc} {\left(\cos\;\alpha \right)}^{2} & {\left(\sin\;\alpha \right)}^{2} & \cos\;\alpha \sin\;\alpha \\ {\left(\cos\;\left(\alpha +\theta \right)\right)}^{2} & {\left(\sin\;\left(\alpha +\theta \right)\right)}^{2} & \sin\;\left(\alpha +\theta \right)\cos\;\left(\alpha +\theta \right)\\ 2\cos\;\alpha \cos\;\left(\alpha +\theta \right) & 2\sin\;\left(\alpha +\theta \right)\sin\;\alpha & \sin\;\alpha \cos\;\left(\alpha +\theta \right)+\cos\;\alpha \sin\;\left(\alpha +\theta \right) \end{array}\right]\left\{\begin{array}{c} {d\varepsilon }_{11}\\ {d\varepsilon }_{22}\\ {d\gamma }_{12} \end{array}\right\}=\left\{d\varepsilon \right\}$$

Define that
(11)$$\left[M_{T}\right]=\left[\begin{array}{ccc} {\left(\cos\;\alpha \right)}^{2} & {\left(\sin\;\alpha \right)}^{2} & \cos\;\alpha \sin\;\alpha \\ {\left(\cos\;\left(\alpha +\theta \right)\right)}^{2} & {\left(\sin\;\left(\alpha +\theta \right)\right)}^{2} & \sin\;\left(\alpha +\theta \right)\cos\;\left(\alpha +\theta \right)\\ 2\cos\;\alpha \cos\;\left(\alpha +\theta \right) & 2\sin\;\left(\alpha +\theta \right)\sin\;\alpha & \sin\;\alpha \cos\;\left(\alpha +\theta \right)+\cos\;\alpha \sin\;\left(\alpha +\theta \right) \end{array}\right]$$

Using Equation (10), the transferring matrix between stress $\left\{d\tilde{\delta }\right\}$ and {dδ}
(12)$$\left\{d\delta \right\}=\left\{\begin{array}{l} {d\delta }_{11}\\ {d\delta }_{22}\\ {d\delta }_{12} \end{array}\right\}={\left[M_{T}\right]}^{T}\left\{\begin{array}{l} d{\tilde{\delta }}_{11}\\ d{\tilde{\delta }}_{22}\\ d{\tilde{\delta }}_{12} \end{array}\right\}=\left\{d\tilde{\delta }\right\}$$

Substitute Equations (1) and (10) into Equation (12) can obtain
(13)$$\left\{d\delta \right\}={\left[M_{T}\right]}^{T}\left[\tilde{D}\left(\varepsilon \right)\right]\left[M_{T}\right]\left\{d\varepsilon \right\}=\left[D\right]\left\{d\varepsilon \right\}$$
where [D] is the elastic matrix of the orthogonal coordinate system. It is easy to obtain the following relationship
(14)$$\left[D\right]={\left[M_{T}\right]}^{T}\left[\tilde{D}\right]\left[M_{T}\right]$$

### 2.2. The Analysis of Viscosity Characteristics of the Resin

Different from dry fabrics, the resin viscosity will change while heating, which affects the prepreg’s stiffness. In order to investigate the influence of resin, this study used the wp-3011 carbon fiber plain-woven thermoset prepreg (Weihai Guangwei composite material Co., Ltd., Weihai, China) as the test material. The material parameters can be found in Table 2. 

Three samples are compared to observe the heating effect. Each sample material is a square prepreg of 50 mm × 50 mm. Before the measurement, an incandescent lamp is used to heat the material, and an infrared thermometer is used to measure the sample temperature. Each experiment is conducted three times, and the sample pictures are shown in Figure 4. With the help of the Keyence VHX-600 digital microscope, the observation shows that at 15 °C, the resin is solidified and dispersed (the solidified resin is distributed in the lowest part, almost not connected to each other), and there is less than half the area uncovered by resin. At 30 °C, the semi-cured resin begins to flow, and the resin-uncovered area begins to decrease. At 45 °C, the completely melted resin is evenly distributed on most areas of the fiber. 

Because the resin of the prepreg is difficult to separate and measure, this article refers to the resin properties of a similar Toray^®^ prepreg. From the datasheet of Toray^®^ G-94M PREPREG SYSTEM, it could be found that before 90 °C, from the data curve of Figure 5, the viscosity and the temperature of the resin show a nearing exponential decline relationship.

From the above observation, it is reasonable to indicate that the temperature affects resin viscosity, which in turn affects yarn sliding ability, and ultimately affects the deformation of CFPWTP. Therefore, the effect of temperature on the viscosity is equivalent to the direct impact of temperature on the CFPWTP deforming ability (stiffness). However, due to the complex interface between fibers, the influence of temperature on CFPWTP seems hard to analyze from the perspective of viscosity alone. In addition, the resin is a polymer material with viscoelastic characteristics. In the same way, temperature also changes its viscoelasticity, which affects the cumulative equivalent elasticity of the CFPWTP. Therefore, the influence of temperature on resin viscoelasticity is also equivalent to the effect of temperature on the CFPWTP deformation ability (stiffness). Moreover, heating affects analysis on CFPWTP in terms of stiffness avoids researching complex contact interfaces. Considering the characteristics, both viscosity and viscoelasticity decrease with heating, and they have similar effects on the deformability of CFPWTP. For simplifying the later analysis, the effect of temperature on the resin is considered entirely on the resin’s viscoelasticity. 

Based on the above analysis, another assumption is created here: fiber and resin are independent. Therefore, the resin viscoelastic modulus can be theoretically added to the fabric model elastic modulus $\left[\tilde{D}\;(\varepsilon )\right]$. Thus, the equivalent Section 2.1 fiber model with the resin characteristics superimposed is considered the model of CFWPTP. The advantage of this method is that the parameter curve of the new CFPWTP model can be obtained through the previous PF and uniaxial tests at specific temperatures. Besides, the thermo-mechanical response envelope surface can then be established with multiple curves at different temperatures, and the fitting equation of the envelope surface is the constitutive equation $\left[\tilde{D}(\varepsilon ,T)\right]$ of the CFPWTP model, where $\left[\tilde{D}(\varepsilon ,T)\right]$ is the revised elastic matrix of Equation (14) $\left[\tilde{D}(\varepsilon )\right]$ with the temperature parameter T. 

### 2.3. Constitutive Model of CFPWTP Material

Because the CFPWTP model is theoretically based on the continuum mechanics. By taking advantage of the differentiable characteristics of the continuum, the $\left[\tilde{D}(\varepsilon ,T)\right]$ could be expanded with the Taylor’s series.
(15)$$\left[\tilde{D}(\varepsilon ,T)\right]=C+C_{11}\left[F\left(\varepsilon \right)\right]+C_{21}\left[F\left(T\right)\right]+C_{12}\left[F\left(\varepsilon^{2}\right)\right]+C_{22}\left[F\left(T^{2}\right)\right]+\cdots$$
where C and C_ij_ are constants (i, j = positive integer), and the strain ε and the temperature T are the independent variables. Similarly, its elastic matrix can be defined as
(16)$$\left[\tilde{D}\right]=\left[{\tilde{D}}_{\mathrm{preg}}\left(\varepsilon ,T\right)\right]=\left[\begin{array}{cc} {\tilde{D}}_{\mathrm{preg}}{}^{11}\left(\varepsilon ,T\right) & {\tilde{D}}_{\mathrm{preg}}{}^{12}\left(\varepsilon ,T\right)\\ \begin{array}{l} {\tilde{D}}_{\mathrm{preg}}{}^{21}\left(\varepsilon ,T\right)\\ 0 \end{array} & \begin{array}{l} {\tilde{D}}_{\mathrm{preg}}{}^{22}\left(\varepsilon ,T\right)\\ 0 \end{array} \end{array}\begin{array}{c} 0\\ 0\\ {\tilde{D}}_{\mathrm{preg}}{}^{33}\left(\varepsilon ,T\right) \end{array}\right]$$

For calculating the material elasticity matrix $\left[{\tilde{D}}_{\mathrm{preg}}\left(\varepsilon ,T\right)\right]$ of the Equation (16), the following method will be adopted here:

(a)Obtain the force-displacement characteristic curve C_F-P_ through the test at a specific temperature, T.(b)Combined with Equations (6)–(9), the characteristic curve C_F-P_ is transformed into the stress-strain curve C_δ-ε_ of the temperature T. (c)Using characteristic curve set {C_δ-ε_(T1), C_δ-ε_(T2), C_δ-ε_(T3), C_δ-ε_(T4)⋯} to construct the surface S_δ-ε-T_ equation, which characterizes the relationship between temperature T, stress δ and strain ε.(d)Based on Equation (15), calculating the fitting surface equation $\left[{\tilde{D}}_{\mathrm{preg}}^{\mathrm{ij}}\left(\varepsilon ,T\right)\right]$ with the nonlinear regression, where i, j = 1, 2, 3.

As for the range of axial tensile curve C_F-P_, the setting range of the path is from 0 to the distance when the material breaks. Similarly, in the range of the PF tests’ curve C_F-P_, the setting range is from 0 to the distance when the device arms contact. As for the limit of temperature T, the setting value is based on the following considerations: it can be seen from Figure 4 that the resin is approximately solid at 15 °C but melts and evenly distributes at 45 °C. This indicates that the resistance of the resin to the fiber is small enough at 45 °C. In addition, the excessive temperature will aggravate the internal chemical action of the resin and lead to the early curing of the material. Combined with the above analysis, the temperature in this study is set in three values: 15 °C, 30 °C and 45 °C.

## 3. Material Characteristic Determination

### 3.1. Test Device for Material Characteristics

In order to obtain the material property for Equation (15), the PF test and axial tensile test are carried out in this section, and the test devices are shown in Figure 3. The test machine is INSTRON 5982-100 kN. The PF test device arm length L_arm_ is 300 mm, and the size of the cross-shaped material is shown in Figure 6. As for the tensile test, the preparation size of materials is 60 mm × 200 mm. The effective tensile length L_mat2_ is 100 mm, as shown in Figure 7. The stretching speed of the test machine is always 10 mm/min. In addition, because the fabric material is relatively soft, the material may relax without tension and damage accuracy. The tightening force (20 N) is applied to ensure the material tightening.

In addition, the initial temperature of the test is 15 °C. Due to the large size of the cross-shaped material, controllable hot airflow is used. Its advantage is that the material is surrounded by hot air, which can maintain a uniform temperature field even if the fabric moves. 

For measuring the heating effect of the PF test, an infrared thermometer is used to detect the temperature values at five points, see Figure 8. The specific values are shown in Table 3. The material temperature error increases as the heating value rises. The error changes from the first 0.1 (15 °C) to the next 0.2 (30 °C) to the last 0.4 (45 °C). It is because the increase in heating leads to easier heat exchange between hot wind and the surrounding air, and it finally leads to an increase in the heat dissipation of materials. In addition, due to the small size of the material in the axial tensile test, only the center area of the test material is measured. The maximum error of the temperature before and after the test at 15 °C, 30 °C and 45 °C are all less than 0.2 °C. Based on the above analysis, the deviation of surface temperature is less than 1 °C. It is supposed that this heating method will not damage test accuracy.

### 3.2. Parameters Calculation and Fitting Approaches

In order to obtain the specific form of Equation (15), tensile tests are carried out in this part to obtain the material characteristic curve. For accuracy and effectiveness, every test is conducted three times, and the average value curves and the error bar are drawn in Figure 9 and Figure 10. 

As for the axial tensile test, the curves can be seen in Figure 9. Firstly, the three curves have similar shapes: 1. pre-tensioning stage (0–0.25 mm); 2. straightening stage (0.025–1 mm); 3 fracture stage (1–2 mm). Although the three curves are measured at different temperatures, there is only a small difference in the tensile strength—680 N at 15 °C; 671 N at 30 °C; and 670 N at 45 °C—indicating that the temperature has little effect on the axial tensile strength. The error value is small (<20 N) in the rising stage but becomes large after a fracture (>100 N). Thus, the curve-fitting process is only applied to the curves before fracture. Through comparison, the max amplitude of the error bar of the three curves is about 20 N, due to the fact that the max error value/maximum value (20 N/670 N) of the curve is less than 3%. In this paper, it is believed that the curve can characterize the axial properties of the material

As for the shearing test, three shear curves show similarity. From Figure 10, on the whole, the three curves are nonlinear and similar to exponential curves. However, with careful analysis, it can be found that the curve contains approximately linear parts (0–100) and fast-rising parts (100–160). From the view of the curves’ maximum value at the linear stage (0–100 mm), the max force of 15 °C is approximately 60 N. At the same stage, the 30 °C ultimate value is about 30 N, while the 45 °C value is 20 N. When the material enters the nonlinear stage (100–160), the maximum value of the 15 °C is about 1600 N, which ranks first, and the value at 30 °C is second at about 790 N. The ultimate value of the 45 °C curves is about 300 N at the third level. Because the maximum curve value (300 N) at 45 °C is 18.7% of the curve value (1600 N) at 15 °C, temperature influence on shear deformation is not negligible during analysis. The above material curve characteristics can be explained from the following aspects. In the first nearing linear parts, the prepregs experienced pure shear condition where the warp and weft tows rotate then compact, resulting in the force increasing gradually. In the nonlinear parts (100–160), the fiber rotation is restricted due to shear locking, and the force along the fiber induces a sharp increase. As for the error, in the rising stage, the max error of the 15 °C curves is 30 N, the max value is 31 N for the 30 °C, and the error amplitude is 20 N for the 45 °C. Because the max error value/maximum (31/790) value of the curve is no more than 7%, and the error is evenly distributed on both sides of the curve, these curves are considered to be sufficient to characterize the shear properties of the material. 

Because the property curves are available, a polynomial fitting operation is performed in this section, and the R^2^ method is used to judge the fitting accuracy. The criterion is that R^2^ closer to 1 means better quality. The calculating equations are below.
(17)$$\left\{ \begin{array}{l} \mathrm{RSS}=\sum_{i=1}^{n}{{(y}_{i}-f_{i})}^{2}\\ \mathrm{TSS}=\sum_{i=1}^{n}{{(y}_{i}-\bar{y})}^{2}\\ R^{2}=1-\frac{\mathrm{RSS}}{\mathrm{TSS}} \end{array}\right.$$
where RSS is the error sum of squares, TSS is the total sum of squares, where n is the number of original data, y_i_ is the i-th data, f_i_ is the i-th predicted value, and $\bar{y}$ is the mean value of total n data. Table 4 shows that the R^2^ values of all the fitted curves are very close to 1, which indicates that the fitted curves can effectively characterize the original curves.

Next, to obtain $\left[{\tilde{D}}_{\mathrm{preg}}\left(\varepsilon ,T\right)\right]$, the following operations are arranged here:

1. Convert the polynomial curve equations at each temperature into a stress-strain relation curve according to the Equations (6)–(9). The curves of the modulus are presented in Figure 11 and Figure 12.

2. Arrange the stress–strain fitting curves in the coordinate system in order of the temperature from low to high. Create an envelope surface for the above three curves. This surface is the stress–strain–temperature characteristic surface of CFPWTP.

3. According to Equation (16), the fitting operation is performed on the envelope surface. The shear and axial characteristic envelope surfaces are shown in Figure 13 and Figure 14. The fitting equations $\left[{\tilde{D}}_{\mathrm{preg}}\left(\varepsilon ,T\right)\right]$ are presented in Equation (18).
(18)$$\left[{\tilde{D}}_{\mathrm{preg}}\left(\varepsilon ,T\right)\right]=\left\{ \begin{array}{l} \left[{\tilde{D}}_{\mathrm{preg}}{}^{11}\left(\varepsilon ,T\right)\right]=34.5-5380\varepsilon -0.63T+56530\varepsilon^{2}+120.8\varepsilon T-0.0055T^{2}=\left[{\tilde{D}}_{\mathrm{preg}}{}^{22}\left(\varepsilon ,T\right)\right]\\ \left[{\tilde{D}}_{\mathrm{preg}}{}^{12}\left(\varepsilon ,T\right)\right]=0.02\times \left[{\tilde{D}}_{\mathrm{preg}}{}^{11}\left(\varepsilon ,T\right)\right]=\left[{\tilde{D}}_{\mathrm{preg}}{}^{21}\left(\varepsilon ,T\right)\right]\\ \left[{\tilde{D}}_{\mathrm{preg}}{}^{33}\left(\gamma ,T\right)\right]=-1.01-0.065\gamma +0.21T-0.0021\gamma^{2}+0.0013\gamma T-0.006T^{2}+\\ \;0.00028\gamma^{3}-0.00051\gamma^{2}T+0.00033\gamma T^{2} \end{array}\right.$$

From Table 4, the R^2^ parameters of the shear surface equation $\left[{\tilde{D}}_{\mathrm{preg}}{}^{33}\left(\gamma ,T\right)\right]$ and axial tensile surface equation $\left[{\tilde{D}}_{\mathrm{preg}}{}^{11}\left(\varepsilon ,T\right)\right]$ are 0.980 and 0.997, respectively. Because the two values are very close to 1, it also indicates that the surface equations have fully included the characteristics of the modulus curves. Besides, the traditional method only tests at one temperature and obtains the fixed elastic modulus of Equation (18). The characteristic modulus obtained in this study is able to change with temperature, which can be used to determine the optimal temperature with broader adaptability.

## 4. Forming Simulation and Experiment

### 4.1. Introduction of Simulation and Test

Because the CFPWTP characteristic matrix $\left[{\tilde{D}}_{\mathrm{preg}}\left(\varepsilon ,T\right)\right]$ is available, this section presents a finite element-forming simulation on the Abaqus CAE 2016 - SIMULA^TM^ by Dassault Systèmes^®^, Paris, France. The simulation model includes four parts: the punch, the cover plate, the CFPWTP material, and the die mold, see Figure 15. Specifically, the model’s element type is S4R and its thickness is 0.2 mm. According to Equation (18), the material properties are written in the VFABRIC subroutines. Besides, the punch is a saddle-shaped mold with double curvature, and the geometric feature of the punch is symmetrical about the X-axis and Y-axis. The top surface is flat and the height of punch is 40 mm, and its side wall is part of a large round barrel (R_2_ = 400 mm). The side wall and bottom surface are transited by round corners (R_4_ = 10 mm). The front and rear end faces of the punch are an arc with a radius of 40 mm. In particular, the distance between the die and the punch is 1.8 mm. The CFPWTP material’s size is 500 mm × 500 mm × 0.2 mm, and the friction coefficient between every two surfaces is set to 0.1. The operation process of simulation is as follows. 1. Place the material on the die in advance. 2. Place the cover plate on the material, where the angle between the material and the X-axis is 45°. 3. Set the temperature at 15 °C, 30 °C, and 45 °C separately. 4. Apply concentrated forces 8 × 50 N on the cover plate and 2000 N on the punch. The simulation is run on ThinkPad P15v i7-2.60 GHz, and the results are given in Figure 15 and Figure 16. 

### 4.2. Forming Simulation and Test

From Figure 15, the deformed material shows an approximated diamond shape from the top, and the material surface without cover plate pressing is full of wrinkles. Because the whole material is symmetrical, only the third quadrant section will be analyzed and discussed.

For quantitative analysis, the length and width of deformed material at three temperatures are presented in Table 5. A comparison of these values shows that in the width direction (X direction), the length of the material increases first and then decreases, but the change is very small (<4 mm), basically maintained at 340 mm at three temperatures. However, in the length direction (Y direction), the size of the material decreases when the temperature rises. Specifically, the initial value is almost unchanged at 353 mm, but it drops sharply to 336.2 mm at 45 °C. In addition, there is an uncovered triangular area at the lower-left corner of the mold after stamping. As the temperature increases, the material’s deformation intensifies, resulting in the uncovered blue area becoming larger. These phenomena are because the deformation difficulty of the CFPWTP decreases with the heating, resulting in easier deformation of the material. 

To further analyze the character of the material, the in-plane shear stress SFABRIC12 of the material is analyzed, as shown in Figure 16. From the stress ranges, it can be seen that when T = 15 °C, the amplitude range of SFABRIC12 is the largest (−5.442 × 10^6^, 1.116 × 10^7^), and the overall distribution is uneven. At 45 °C, the amplitude range of SFABRIC12 is the smallest, the overall distribution is uniform, and the average value is about (−1.667 × 10^5^, 1.667 × 10^5^). The value range at 30 °C is (−6.97 × 10^5^, 5.37 × 10^5^) which is between them, and the stress distribution is more uniform than at 15 °C. From the point of view of stress distribution, the stress of the whole surface presents a uniform state at 45 °C. However, at 30 °C, the maximum positive shear stress and the maximum negative shear stress are mainly near the position where the cover plate is pressed. Large stress is much more concentrated at the lower left side of the die mold. At 15 °C, the whole surface shear stress changes sharply, and the shear stress range enlarges significantly. This shows that as the temperature decreases, folds will first appear near the pressed area of the cover plate. As the temperature decreases, the material deformation becomes more difficult, and the folds will further spread to the whole surface. From the above analysis, it can be concluded that the shear stress of the material decreases and tends to be uniform with the temperature rise.

### 4.3. Forming Test and Comparison

To examine the simulation accuracy, the forming test with the 500 mm × 500 mm wp-3011 prepreg is arranged in this part. The punch, cover plate and die mold are made of aluminum alloy. The punch pressing force is 2000 N, and the cover plate is applied with four air cylinders (4 × 100 N). Besides, the die mold is equipped with a temperature controller for preheating purposes. The resin’s melting could affect the friction coefficient between the material and the punch and the die surface. For preventing the unpleasant friction coefficient changes, the whole surfaces of the punch, the die and the cover plate are pasted with the smooth Teflon coating to stabilize the friction. Then the test results are as follows.

By drawing a grid on the photos, the size of the material is measured, as shown in Figure 17. The size errors of the simulation and test are calculated, and the data are shown in Table 5 and Figure 18. It can be seen that the material length in the X -direction will increase first and then decrease while heating, but the magnitude (about 342 mm) of the change is small (<4 mm). As for the material length in the Y-direction, the length first decreases slightly from 346.4 mm at 15 °C to 343.4 mm at 30 °C and then decreases substantially from 30 °C to 353.5 mm at 45 °C. In addition, overall, the X_err_ is generally smaller than Y_err_, in which the maximum error in the X direction is 0.9%, while the maximum error in the Y direction is 5.1%. The possible reason for simulation and test difference is that the mold shape makes the material easier to shrink in the Y direction, so the deformation in the Y direction is more significant, bringing more error. As for the shape, it can be seen from Figure 15 that as the temperature rises, the uncovered area at the lower-left corner of the die becomes larger, which is consistent with the trend predicted by the simulations. Finally, by comparing the test results and simulation, it can be seen from Figure 18 that although there are shape deviations, the maximum error is not more than 5.1%, and the simulation and test results show the same deforming trend. Therefore, the model established in this paper can still meet the needs of following defect prediction.

### 4.4. Wrinkles Prediction and Comparison

The typical defect of woven composites is the wrinkle. In order to predict the possible defect location with the established model, two criteria of the wrinkles are established. 1. According to the reference [13], the maximum shear angle should not exceed the shear locking angle of 45° (in-plain wrinkle). 2. The normal distance from the mesh node to the die surface should be equal to the material thickness (out-of-plain wrinkles). 

Through quantitative analysis, Figure 19 shows that at 15 °C, the shear angle of every element is below the shear lock angle, but there are obvious folds whose max normal height is about 2 mm. This type of fold belongs to out-of-plane bending, see red arrows. At 30 °C, the material has in-plane and out-of-plane folds. Among them, the most serious out-of-plane fold height is 0.8 mm, and the maximum shear angle of the in-plane fold is 51 °. At 45 °C, the element fluctuation is lower than that of the 30 °C conditions, and the amplitude is less than 0.4 mm. The largest shear angle is 29°. Moreover, as for wrinkles distribution, at 15 °C, folds are mainly distributed at the corner of the front end of the die and the transition area of the side wall. At 30 °C, the out of plane folds are distributed in the transition region, and the in-plane folds are distributed at the left top edge of the mold. At 45 °C, the whole surface is smooth, but there are only small folds at the sharp corners of the mold. Compared with the shear stress distribution of SFABRIC12, the wrinkles’ location shows consistency with the large shear stress position on the mold.

The above phenomenon happens because of a few reasons. 1. The geometric constraint: the press of the cover plate makes it difficult for the material at this position to bulge substantially, so the folds at this position are generally in-plane folds. In the transition region of the sidewall, the material is in a free state, and the material will turn 90° in this region, which leads to the region being in a bending state all the time, and it is very easy to produce out of plane folds. 2. The temperature effect: the temperature affects the shear deformation stiffness of materials. When the temperature is high enough, e.g., 45 °C, the shear stiffness and shear stress will decrease, both of which are conducive to the smooth and large deformation of materials without folds. When the temperature is not high enough, e.g., 30 °C, the material may have both in-plane and out-of-plane folds, depending on the situation and constraint of the material. At low temperature, e.g., 15 °C, the material is difficult to deform. At this time, the internal stress of the material needs to be released by the deformation which is prone to causing the out-of-plane defect.

From the test results, Figure 20 shows that at 15 °C, there are two big wrinkles on the side wall with a height of 5 mm (see red arrows). According to Figure 15, there is also some waviness at the outer edge of the mold with a height of 1 mm. At 30 °C, the side wall folds are reduced to one, and the height is reduced to 3 mm. The waviness in the outer edge of the mold area is aggravated, but the height is less than 1 mm, as shown in Figure 15. At 45 °C, the original folds shrink, and a new small wrinkle in the corner appears, and the height of it is about 1 mm. The waviness in the top cover plate area also disappears. The big folds are mainly distributed in the outer space of the mold. 

By comparing the simulation and test results, the simulation is consistent with the test in fold position and deformation trend under different temperatures. It is considered that the prediction effectiveness has been verified. However, it can be found that there is a certain deviation between the model and the test, such as the model failing to predict the waviness at the outer edge of the mold at 15 °C. It is because firstly, in order to simplify the model, only three temperatures are considered in the model; secondly, the coupling effect inside the material is ignored in the model. To decrease the above deviation, the model will add more parameters in future work. 

## 5. Conclusions

The fabric material deforming ability is crucial for composite aerospace part manufacturing. An equivalent continuum model is presented for predicting the mechanics of CFPWTP material. Based on the analysis of materials, this paper arranges mechanics tests at 15 °C, 30 °C and 45 °C. By Taylor’s expansion formula and the surface fitting method, the surface constitutive equation of the material is obtained. Besides, a forming test is carried out to examine the model’s effectiveness. Results prove that 1., the increase in temperature can reduce the deformation stiffness of the material and effectively inhibit the folding of the CFPWTP; 2., the CFPWTP model in this paper contains the temperature characteristic of the resin and has high accuracy—the width error is less than 0.9%, and the length error is less than 5.1%; 3., the fold position predicted by simulation is basically similar to the test results. The models and methods in this paper can provide effective tools for designers. In addition, to make further improvement to the model accuracy, the model will introduce more parameters such as parameter coupling in the follow-up study. 

## Figures and Tables

**Figure 1 polymers-14-02618-f001:**
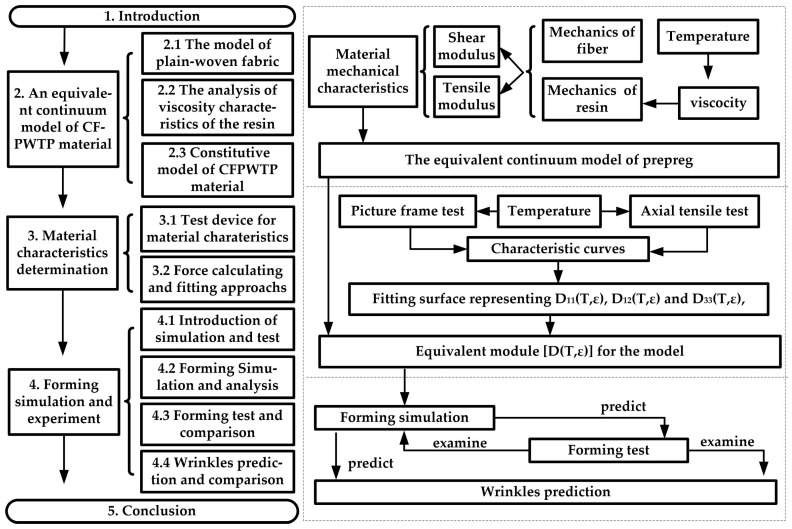
The architecture of the paper includes the model construction and validation approaches.

**Figure 2 polymers-14-02618-f002:**
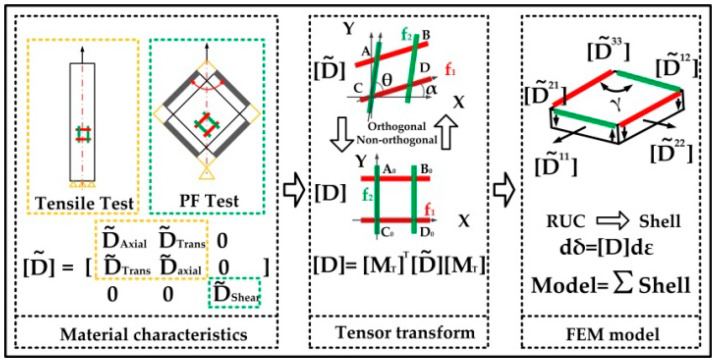
The modeling process.

**Figure 3 polymers-14-02618-f003:**
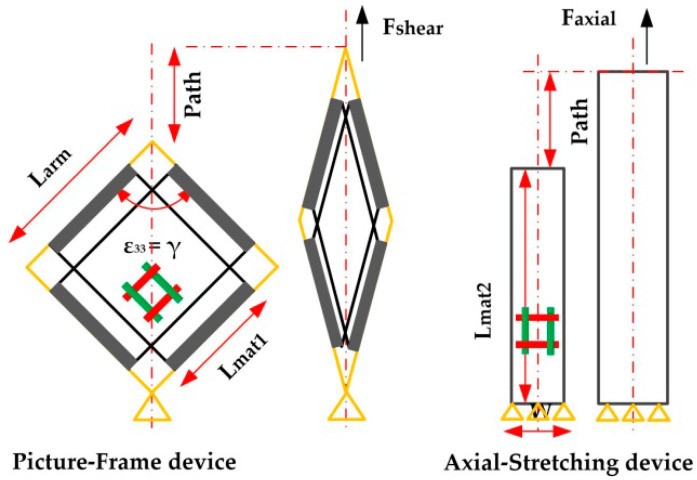
Schematic diagram of PF device and axial tensile device.

**Figure 4 polymers-14-02618-f004:**
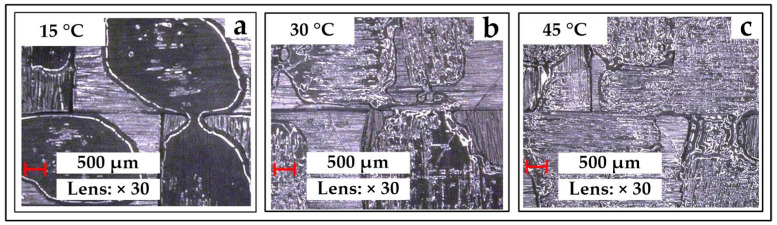
The resin conditions at three temperatures with magnification of the microscope is 30: (**a**) the condition at 15 °C; (**b**) the condition at 30 °C; (**c**) the condition at 45 °C.

**Figure 5 polymers-14-02618-f005:**
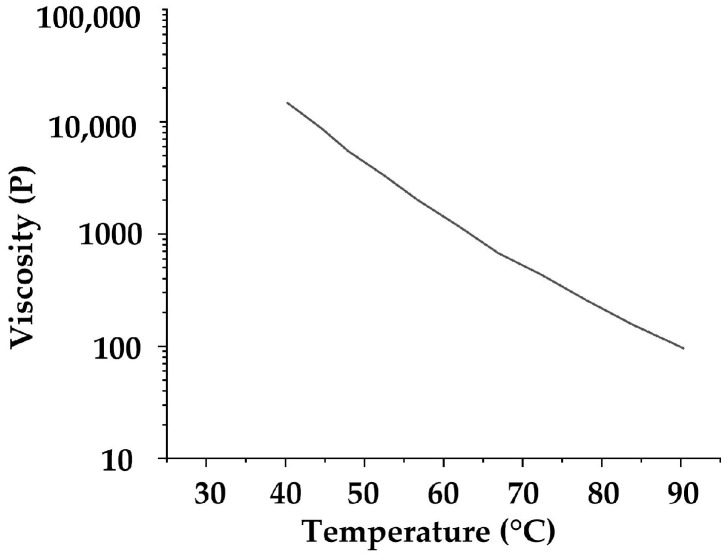
The relationship between resin viscosity and temperature in the exponential coordinate [50].

**Figure 6 polymers-14-02618-f006:**
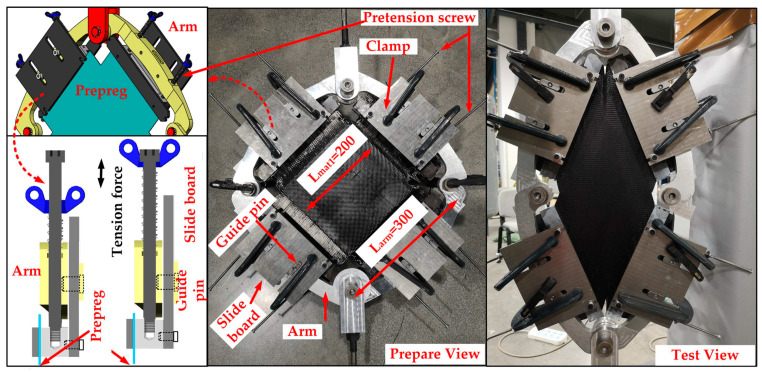
Picture of the PF test device.

**Figure 7 polymers-14-02618-f007:**
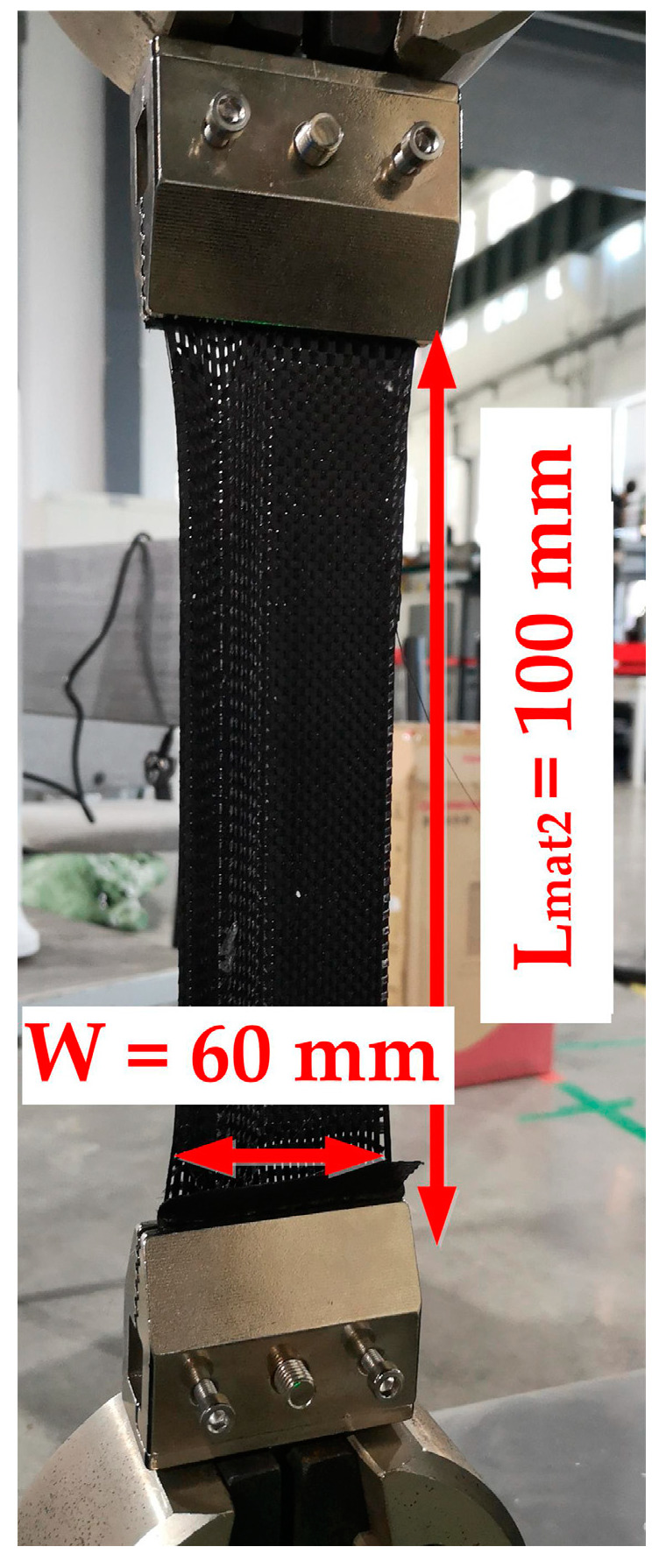
Size of the axial tensile material.

**Figure 8 polymers-14-02618-f008:**
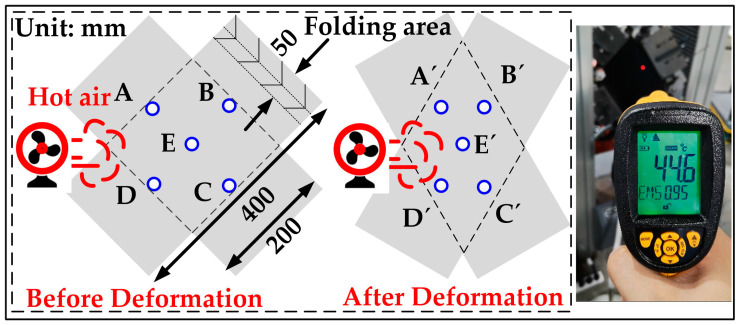
Temperature detection points on material.

**Figure 9 polymers-14-02618-f009:**
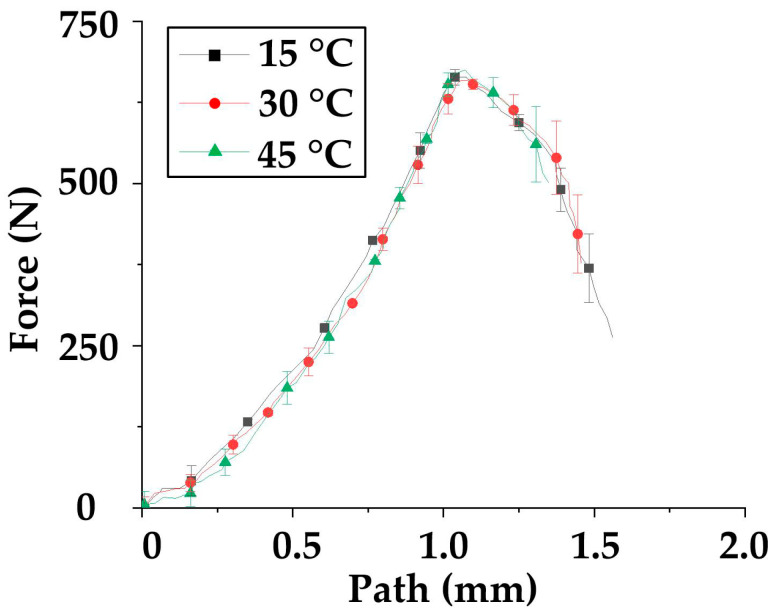
The axial tensile test curves.

**Figure 10 polymers-14-02618-f010:**
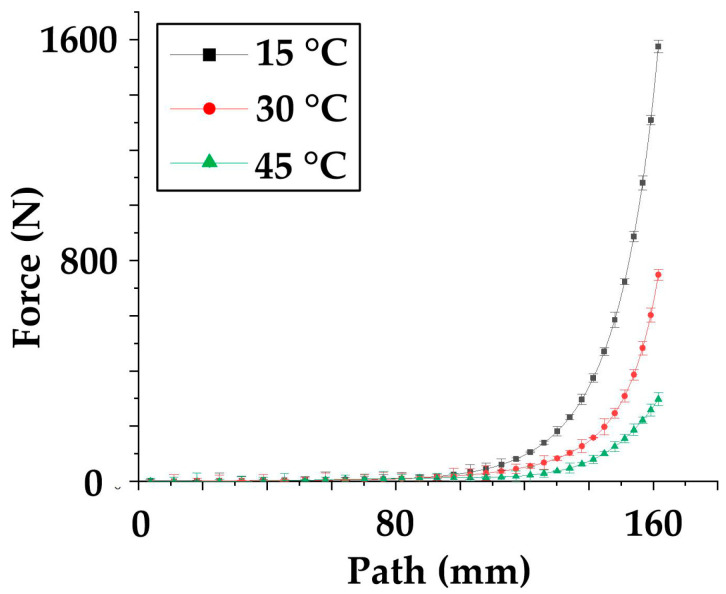
The PF test curves.

**Figure 11 polymers-14-02618-f011:**
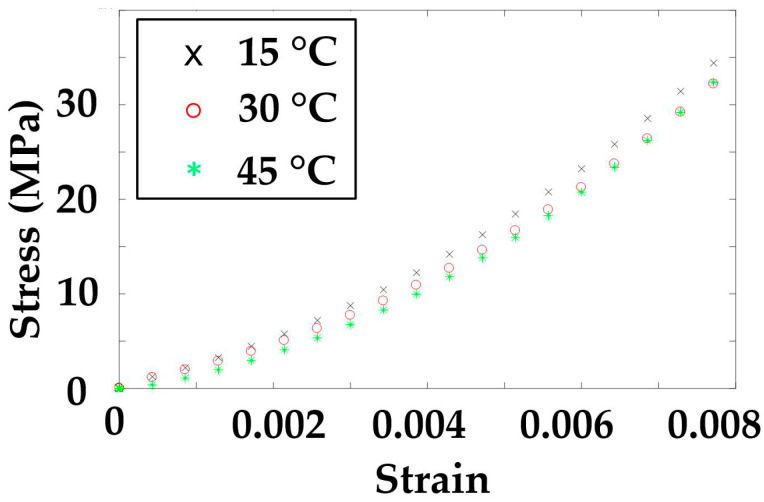
Axial stress–strain curves at three temperatures.

**Figure 12 polymers-14-02618-f012:**
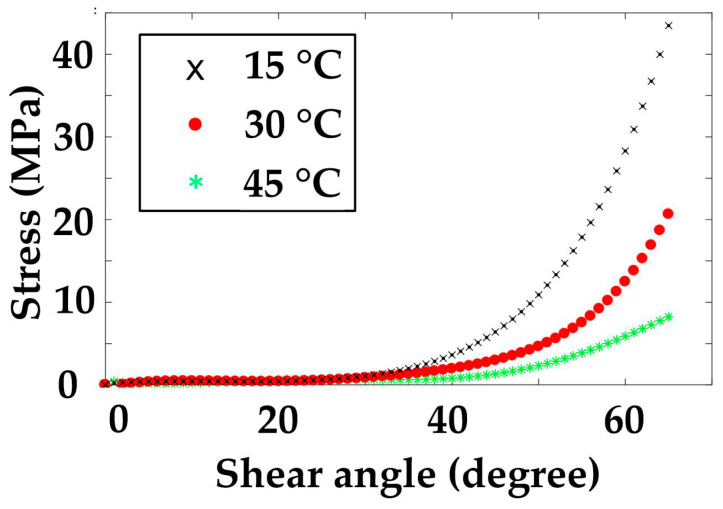
Shear stress–strain curves at three temperatures.

**Figure 13 polymers-14-02618-f013:**
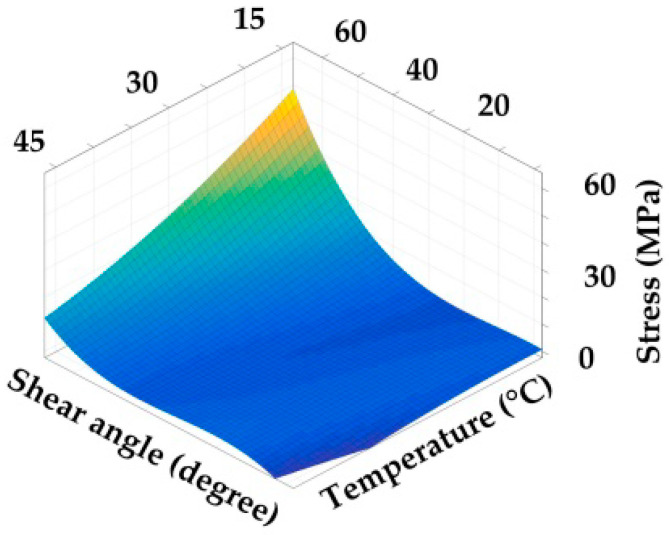
Stress–strain–temperature envelope surface of shear test.

**Figure 14 polymers-14-02618-f014:**
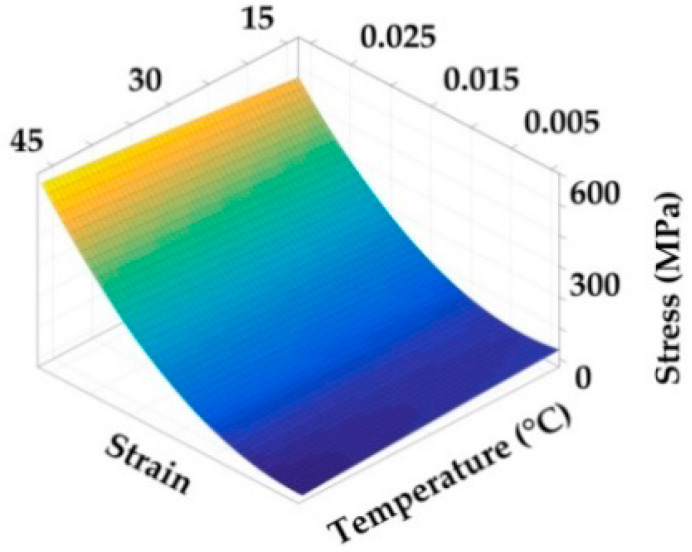
Stress–strain–temperature envelope surface of axial tensile test.

**Figure 15 polymers-14-02618-f015:**
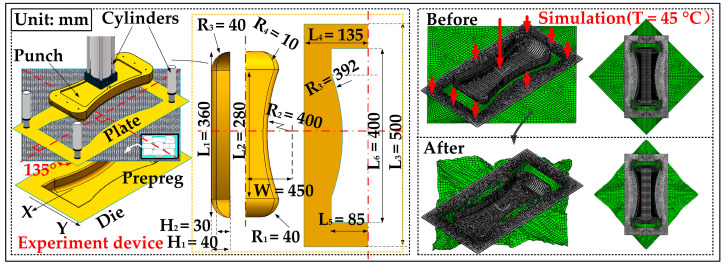
Schematic diagram of stamping device and simulation.

**Figure 16 polymers-14-02618-f016:**
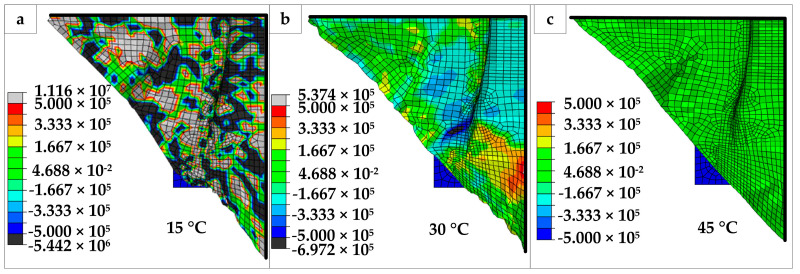
The shear stress distribution at different temperatures: (**a**) the top view of material at 15 °C; (**b**) the top view of material at 30 °C; (**c**) the top view of material at 45 °C.

**Figure 17 polymers-14-02618-f017:**
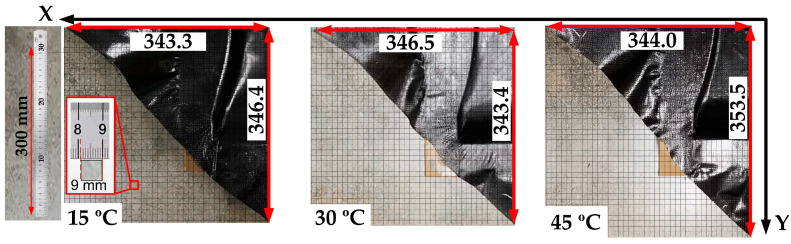
The forming test results at 15 °C, 30 °C and 45 °C.

**Figure 18 polymers-14-02618-f018:**
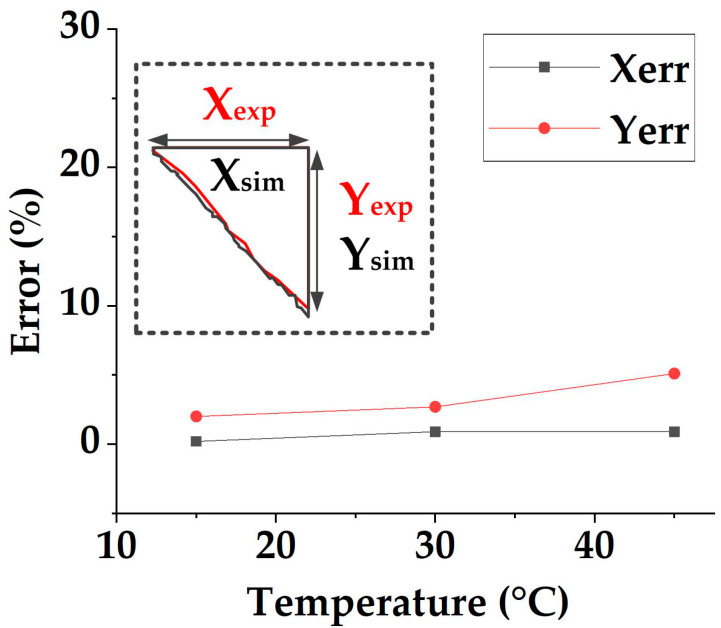
The size comparison of the simulations and tests.

**Figure 19 polymers-14-02618-f019:**
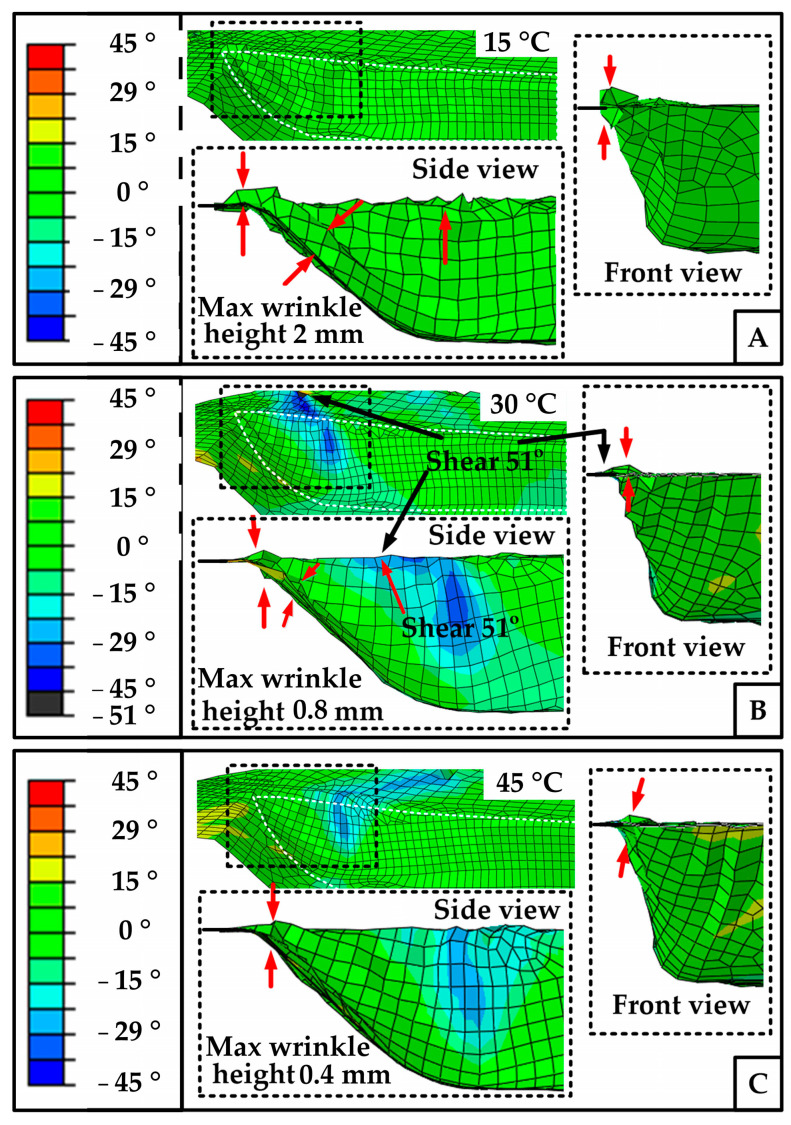
The wrinkle distribution of simulations: (**A**) the side view and front view of corner material at 15 °C; (**B**) the side view and frontview of corner material at 30 °C; (**C**) the side view and front view of corner material at 45 °C.

**Figure 20 polymers-14-02618-f020:**
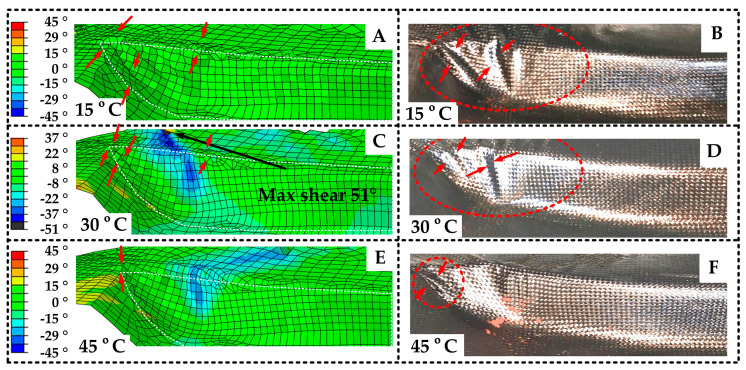
Wrinkles distribution comparisons: (**A**) the wrinkles of simulation at 15 °C; (**B**) the wrinkles of test at 15 °C; (**C**) the wrinkles of simulation at 30 °C; (**D**) the wrinkles of test at 30 °C; (**E**) the wrinkles of simulation at 45 °C; (**F**) the wrinkles of test at 45 °C.

**Table 1 polymers-14-02618-t001:** The parameters of Toray^®^ T1100G carbon fiber and prepreg [45].

Name	Type	Tensile Modulus		Shear Modulus	Thickness
filament	T1100G	324 GPa		Φ = 5 μm	
preprep	Plain Weave (3960 Resin/12K)	0° Tensile Modulus	88.79 GPa	In Plain Shear 3.7 GPa	0.2 mm
		90° Tensile Modulus	88.74 GPa		

**Table 2 polymers-14-02618-t002:** The parameters of test material [49].

Name	Type	Filament Tensile Modulus	Matrix	Thickness
wp-3011	plain weave (TZ300/3K)	230 GPa	epoxy resin	0.2 mm

**Table 3 polymers-14-02618-t003:** The value of temperature detection points.

Set-Value (°C)		Real-Temperature at Point					Max-Diff
		A	B	C	D	E	
15	Before	15.0	15.1	15.1	15.1	15.0	0.1
	After	15.0	15.0	15.0	15.0	14.9	0.1
30	Before	30.1	30.1	30.1	30.0	30.0	0.1
	After	30.1	29.9	29.9	30.0	30.0	0.2
45	Before	45.1	45.1	45.2	45.1	45.0	0.2
	After	45.0	44.6	45.0	45.0	44.8	0.4

**Table 4 polymers-14-02618-t004:** The goodness of fit test.

Title	Shearing Test				Axial Tensile Test			
Curve	15 °C	30 °C	45 °C	$\left[{\tilde{D}}_{\mathrm{prep}}^{33}\left(\varepsilon ,T\right)\right]$	15 °C	30 °C	45 °C	$\left[{\tilde{D}}_{\mathrm{prep}}^{11}\left(\varepsilon ,T\right)\right]$
R²	0.998	0.993	0.995	0.980	0.999	0.999	0.997	0.997

**Table 5 polymers-14-02618-t005:** Comparison error between simulation and experiment.

Size Error (mm)	15 °C	30 °C	45 °C
X_sim_	342.5	343.3	340.7
X_exp_	343.3	346.5	344.0
X_err_ = (X_exp_ − X_sim_)/X_exp_	0.2%	0.9%	0.9%
Y_sim_	353.5	353.2	336.2
Y_exp_	346.4	343.4	353.5
Y_err_ = (Y_exp_ − X_sim_)/Y_exp_	2.0%	2.7%	5.1%

## Data Availability

The data presented in this study are available on request from the corresponding author.

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
