# Peer review of "A Mechanics Analysis of Carbon Fiber Plain-Woven Thermoset Prepreg during Forming Process Considering Temperature Effect"

_polymers, 2022, doi:10.3390/polym14132618_

Round 1
Reviewer 1 Report
Paper on "A mechanics analysis of carbon fiber plain woven thermoset prepreg during forming process considering temperature effect". The paper considers good work, however, many sections need improvement as required:
1- Abstract: line " axial tensile tests at 15/30/45°C are ..." please make " at 15°C, 30°C and 45°C..."
line " Consequently, a saddle-shaped forming simulation
is carried out with the above equations, ..." which equation?? give info..
line" And the errors of predicted shape are 3.43% of width direction and 8.59% of length direction separately". please give more clarification that show the significance of these width and length, must add .
Introduction must be improved and add new 10 references from 2018-2022 to show the research gap. can not count on old references only. [This section must improve it. ]
Many parts of the introduction should delete [; ] and use [.].
line : There has already been much research in fabrics, but few models consider the effect of temperature on the CFPWTP materials[15].." must add reference after " much research in fabrics, " and can not use an old ref. [15] in 2011 to show the research gap. must use new updated ref. [2018-2022], and delete ref. no.[15].
Originality / Novelty is Required.
figure no. 1,2 and 3 , please give ref. ?
section "Experiment and discussion", this section only show the results and analysis but not give and discussion in regards the previous studies and modulus??? must compare the current data with old studies and must give strong discussion to show the novelty of the current study. This section must improve it.
Conclusion: this section the first 2 points are general, please make it more specific to your results and achievements.
It is highly required to add future recommendations.
English needs improvement.
Thanks and All the best.
Reviewer 2 Report
The subject of the paper is of interest and the analysis performed is useful. But unfortunately, the document is not well written and poorly presented. There are several flaws to address before even trying to understand the scientific part of the work. Several important details are missing and it is hard to follow the work done, why it was done like this and most importantly what is the objective and the meaning of the results. Here is a long list of corrections to make.
The Journal’s template was not used to submit the paper.
The written English must be improved.
Use only one decimal to report %.
Add space before the ref. # [x]. Report them using the same format.
Revise how to report authors’ names in the text.
Several spaces are missing throughout the manuscript.
The objective of the paper can be better described : be more specific and specify the parameters studied with their range. State what was done exactly.
Revise the order of presentation : items must be presented in a reasonable series of events/steps.
Do not use « What’s more » !
Always put a space between values and units.
Page numbers would have been helpful.
The sample preparation is not reported : which formulations were used : only one from the company ? How many repetition for each testing conditions ?
Revise everywhere the presentation of temperature (uniformity).
The paper is not very quantitative, report more on the values and compare them.
Figure 1. more details in the caption.
Figure 3 : the caption dies not make sense.
Figure 4; where are these data coming from (nothing said on the measurements). There is no vertical axis title nor units. If the system is reactive, remove the data at high T : i.e. cut at 90oC here…
For mechanical testing, report the type of machine used and the load cell capacity.
Page 6 : PE or PF ?
Figure 6 : this is an image !
Table 1 : report the units.
Table 2 : define « A » and « S ».
Figures 10-11 : complete word for « Temperature ».
Page 10 and following : improve the presentation of the values. Always put units.
Figures 13 and 14 : the caption is not precise : what is presented here ?
Figure 15 : what do we see in these images ?
It would have been interesting to investigate the effect of deformation rate or any other parameter of interest for this system…
References
[4] is incomplete
[7] names ???
[17] names ???
[19] incomplete
[20] no capitals
[25] date missing
Most are not to the required format: mainly the journal and author names…
Reviewer 3 Report
This manuscript introduced an equivalent continuum mechanics model to predict the deformation of carbon fiber plain-woven thermoset prepreg (CFPWTP). Picture frame tests (PF test) and axial tensile tests were utilized to obtain the essential characteristics for the material’s equation. A saddle-shaped forming simulation was carried out based on those equations and an experimental, forming test was also used to compare with the simulation data. The overall research strategy is clear and straightforward. But one of the comments from the reviewer is revising the paper structure. Your strategy should be outlined clearly with a proper structure such as introduction, equivalent model, experiment/methods, result (with discussion), and conclusion. What’s more, some unnecessary data and claims should be avoided to distract the core statements; in the revised manuscript, please add the line numbers for the whole draft. It will be much easier for reviewers to locate the comments and point out the corresponding suggestions. The following revision comments are indispensable/strongly recommended to add or revise later for the publication of this manuscript in Polymers
- Overall, please add the line numbers for the whole manuscript
- You should keep the format of reference consistent. You have two different reference formats among the contexts;
- At the end of Page 2, the authors mentioned “ 2) Because the material is laminated. The thickness can be neglected;” What’s the justification behind this assumption? And in your experiment, the sample thickness (T) is about 50mm, and the length (L)/width (W) is 200mm. It’s hard to neglect the impact from thickness, considering the ratio of T/L is so large.
- On Page 4, the authors mentioned: “the [D(ε)] could be separated: the shear part and the axial part…” How to combine the shear part and axial part later?
- On Page 4 & 5, suggest using W to replace Lmat1 when you refer to the width. Similarly, the T is recommended to replace H for the thickness. And also L mat 2 is the original length of the sample?
- On Page 5, about Fig 3, what characterization method did you use to obtain the figure? Explain it in the contexts or captions. Also, what do you mean by saying “the resin is dispersed at 15C”?
- On Page 6, the authors claimed that “In addition, due to the previous assumption that the prepreg's warp and weft tensile properties are the same …” 1) do you have the tensile result to support this assumption? It should be not hard to demonstrate before you set this hypothesis; 2) On Page 10, you also stated that the CFPWTP’s property is inhomogeneous and anisotropic, does this statement contradict your previous assumption?
- For the Part 3. Experiment and discussion, you’re suggested to separate the experiment method introduction (description) with the following result and discussions, as most of the published papers done. Your part 3 is too complicated with so many details and explanations.
- In the section Part 3.1, you have a typo “PE test device”
- On Page 6, the fixture’s width, you may want to show it in Fig 3, if it’s not strongly related to your result. It’s not necessary to mention all the details, especially when this information is not related to your results, the same story as the clamp area. No need to disclose them if they’re not related.
- For the fixing material, the 50X200 folded area disclosed the thickness and width information. That’s why the reviewer doubts if the thickness can be neglected.
- On Page 8, Section 3.2, you wrote “However, compared with the stiffness of axial fiber, the resin's stiffness is too small, which leads to the promotion of axial elongation and decrease of max force effect by rising the temperature are very limited.” what's your point of this statement?
- On Page 8, Section 3.2, you wrote “…the deformation of the material at low temperature is not the smallest…” 15C has the shortest path deformation for both tensile 7 PF tests. Why do you have such a claim?
- In the same paragraph, you claimed that “CFPWTP’s property is anisotropic”. I assume it should have no further impact on the PF test if this claim is true. However, the tensile test should be impacted in two different directions. How do you select the direction and have you compared the tensile data collected from orthogonal directions?
- Figure 15: Could you explain Fig 15 with more details in the caption? What do you want to illustrate in these comparisons? Do you have the supplementary video to illustrate these tests? And also, please mark/note the key components in those three figures.
- Forming test: Introduce the details about the forming test in the experimental part, as what you did for PF tests and axial tensile tests.
Round 2
Reviewer 1 Report
Paper is accepted after the improvement. Thanks
Reviewer 2 Report
The paper was substantially improved. Better presentation, better written English quality and more details in the discussion. The paper can be published after some minor corrections.
There is still several missing spaces, especially before references “[]”. Revise the whole document.
The caption for Figure 1 can be improved.
Still missing space between values and units.
Figure 4 caption : change “of” by “at”.
Figure 5 caption : “The resin viscosity as a function of temperature.”
Most mechanical figures : MPa
Page 20: several values are missing units (improve presentation also).
